# Performance of a Low-Cost Sensor Community Air Monitoring Network in Imperial County, CA

**DOI:** 10.3390/s20113031

**Published:** 2020-05-27

**Authors:** Paul English, Heather Amato, Esther Bejarano, Graeme Carvlin, Humberto Lugo, Michael Jerrett, Galatea King, Daniel Madrigal, Dan Meltzer, Amanda Northcross, Luis Olmedo, Edmund Seto, Christian Torres, Alexa Wilkie, Michelle Wong

**Affiliations:** 1Dept. of Public Health, Richmond, CA 94804, USA; 2Tracking California, Public Health Institute, Oakland, CA 94607, USA; heather_amato@berkeley.edu (H.A.); galaking22@gmail.com (G.K.); daniel.madrigal@cdph.ca.gov (D.M.); DanMeltzer@gmail.com (D.M.); alexa.wilkie@cdph.ca.gov (A.W.); michelle.wong@cdph.ca.gov (M.W.); 3Comite Civico Del Valle, Brawley, CA 92227, USA; esther@ccvhealth.org (E.B.); betomtz.lugo@gmail.com (H.L.); luis@ccvhealth.org (L.O.); christian@ccvhealth.org (C.T.); 4Environmental and Occupational Health Sciences, University of Washington, Seattle, WA 98195, USA; gncarvlin@gmail.com (G.C.); eseto@uw.edu (E.S.); 5Department of Environmental Health Sciences, School of Public Health, University of California, Los Angeles, CA 90097, USA; mjerrett@ucla.edu; 6Department of Environmental and Occupational Health, George Washington University, Washington, DC 20037, USA; northcross@email.gwu.edu

**Keywords:** low-cost monitors, particulate matter, participatory research

## Abstract

Air monitoring networks developed by communities have potential to reduce exposures and affect environmental health policy, yet there have been few performance evaluations of networks of these sensors in the field. We developed a network of over 40 air sensors in Imperial County, CA, which is delivering real-time data to local communities on levels of particulate matter. We report here on the performance of the Network to date by comparing the low-cost sensor readings to regulatory monitors for 4 years of operation (2015–2018) on a network-wide basis. Annual mean levels of PM_10_ did not differ statistically from regulatory annual means, but did for PM_2.5_ for two out of the 4 years. R^2^s from ordinary least square regression results ranged from 0.16 to 0.67 for PM_10_, and increased each year of operation. Sensor variability was higher among the Network monitors than the regulatory monitors. The Network identified a larger number of pollution episodes and identified under-reporting by the regulatory monitors. The participatory approach of the project resulted in increased engagement from local and state agencies and increased local knowledge about air quality, data interpretation, and health impacts. Community air monitoring networks have the potential to provide real-time reliable data to local populations.

## 1. Introduction

Small, low-cost air quality sensors of improving quality are becoming increasing available, and many public health and research projects are now employing these next-generation air monitors to conduct personal and local-level air monitoring for exposure control [1,2]. This new technology holds great potential to address gaps in regulatory air monitoring data to better characterize air quality at the community level. With the stall of international agreements to decrease emissions to address climate change, local community networks could result in cost-effective improvements to air monitoring and emission reduction at the local scale in the near-term and have policy relevance and positive developments globally for climate change and air pollution in the long-term. Starting in 2014, the Imperial County Community Air Monitoring Network (“the Network”) has combined community participatory and scientific methods to develop a network of over 40 air sensors throughout the county, delivering real-time data to local communities on levels of particulate matter. The Network now has 4 years of data available for performance evaluation. Although the use of these sensors is increasing, there have been few performance evaluations of low-cost networks operating in the field [3,4] and none which have reported with such extensive data. Duvall et al. [3] evaluated the performance of low-cost sensors measuring NO_2_ and ozone in two U.S. cities, and found good agreement of their sensors with reference sites. Mailings et al. [4] developed a low-cost gas sensor and evaluated multiple statistical methods for evaluating sensor performance. In Salerno, Italy, Sofia et al. [5] evaluated the performance of a low-cost network measuring PM_2.5_ and found good agreement of their sensors with gravimetric sampling. Although there have been calibration/validation studies of these sensors, to our knowledge no studies have been conducted evaluating performance of a low-cost network post implementation on a network-wide basis, compared to regulatory data. Further, none of the previous evaluations have been done on community-based air monitoring systems where the community has taken the lead in monitor siting and hardware/software maintenance of the network.

We have previously reported on the monitor calibration/validation [6], and the process for monitor siting [7]. The objective of this paper is to evaluate the performance of the Network to date by comparing the low-cost sensor readings to regulatory monitors for 4 years of operation (2015–2018) on a network/domain-wide basis. We also discuss how the participatory research approach affected the project overall, and the policy and public health actions which have resulted from the project.

## 2. Materials and Methods

Imperial County, CA, is home to a primarily Latino population (84%) and has some of the highest rates of unemployment (47%) and poverty (24%) in the nation [8]. The county is primarily a desert ecosystem, much of which has been converted to agricultural land. The county has a range of air pollution sources that contribute to regular and sustained exceedances of the California PM standards [9,10], including nearly 8 million vehicles annually crossing the U.S.-Mexico border in Calexico [11], an average of about 28,000 acres of agricultural field burned annually [12], and the drying Salton Sea [13].

The Imperial County Community Air Monitoring Network (the Network) was formed by a collaborative group of community, academic, nongovernmental, and government partners designed to fill the need for more detailed data on particulate matter in an area that often exceeds air quality standards. The Network employed a community-based environmental monitoring process in which community members and researchers had specific, well-defined roles as part of an equitable partnership that also includes shared decision-making to determine study direction, plan research protocols, and conduct project activities. Community members played key roles in determining study design, siting and deploying monitors, and data collection. The Network is now producing real-time particulate matter data from 42 low-cost sensors throughout the county. The Network is one of the largest community-based air monitoring networks in the U.S. and, to our knowledge, is the first community-designed network of its size in the world.

The project partners included Comite Civico del Valle (CCV), a community-based organization in Imperial County; Tracking California, a program of the nongovernmental Public Health Institute in collaboration with the CA Department of Public Health; and the Seto research group at the University of Washington (UW). Faculty at the University of California at Los Angeles and George Washington University served in an advisory capacity. The distinct roles of the partners and the initial community engagement structure have been described in detail elsewhere [14].

The monitor selected for the Network was a modified laser-based optical counter (Dylos DC1700, Dylos Corpration, Riverside, CA, USA). The firmware was changed to increase the number of particle size bins from two to four (>0.5, >1.0, >2.5, and >10 μm). Particle counts were converted to mass concentrations to align with health recommendations that are usually based on the latter. Algorithms to convert counts to mass were developed, taking into account relative humidity, based on co-location of the instruments with federal equivalent method (FEM), beta-attenuation monitors (BAMs), and federal reference method (FRM) gravimetric filters. The algorithm, process for the calibration, and field validation, and further specifications of the Dylos monitor have been described in detail elsewhere [6].

The monitor system included the Dylos particle sensor, temperature and relative humidity sensors, a heater and fan, and a microcontroller to allow wireless real-time data transfer to the Internet. Data are sampled once every 10 s from the Dylos and a HIH6130 temperature and humidity sensor. Power consumption is about 3 W without the fan or heater (temp above 120 F), and 203 W with the heater and no fan (temp below 40 F). The monitor components were housed in a NEMA-6 rated enclosure with a cooling fan to sustain optimal sensor performance under Imperial County’s harsh summer conditions. The rationale and siting for the first 40 sensors were based on a two stage process where the community selected the locations of the first 20 sensors primarily based on locations of vulnerable populations and the second 20 based on specifications to meet the spatial requirements for land use regression [7]. We developed a process to involve community residents to select monitoring sites, which involved the use of mobile devices equipped with a custom-designed mobile web form, modified from IVAN Imperial (https://ivan-imperial.org/air), CCV’s existing community environmental reporting website [7]. Residents used government-based monitor siting criteria (such as building height, security, likelihood of available Wi-Fi and AC power supply, and locations of nearby air pollution sources) in their assessment.

The air quality data are transmitted in real-time to data servers managed by CCV once every 10 s (data are averaged every 5 min), converted from particle counts to mass concentrations, checked for quality assurance/quality control, and then fed to the IVAN website. Quality control procedures include flagging unusually low values, data completeness, and routine manual inspection of the sensors. The IVAN website developed by CCV allows public access to real-time data from the Network. The U.S. EPA Air Quality Index (AQI) is calculated for five major air pollutants regulated by the Clean Air Act: ozone, particulate matter, carbon monoxide, sulfur dioxide, and nitrogen dioxide [15]. As this project was only measuring particulate matter, another index was needed. To address this, the particle mass results (in μg/m^3^) were averaged using the U.S. EPA NowCast method [16]. The NowCast method produces a value for PM_10_ and PM_2.5_, which is an average of the previous 12 h. If the air quality that day is stable, then the hours are weighted more evenly (approaching a 12-h average). If the air quality that day is changing, then recent hours are weighted more heavily (approaching an average of the most recent 3-h). The resulting NowCast value (in μg/m^3^) is then converted to a Community Air-Quality Level (CAL)—an indicator developed specifically for this Network with community input—using the U.S. EPA Air Quality Index calculation method. After consultation with the Imperial Community Steering Committee on how to make the data most understandable and useable, CALs were categorized into four categories: green/low risk (0–50), yellow/moderate risk (51–100), orange/unhealthy for sensitive groups (101–150), and red/unhealthy (above 150). The CALs are calculated for each monitor based on current concentrations, with the number and category updated on this website every 5 min. The 30-day and 90-day summary statistics for CALs are calculated using 24-h AQIs.

To compare the overall daily averages of PM_10_ and PM_2.5_ from the Network versus regulatory monitors, we used reference monitor data available from U.S. EPA’s Air Quality System (https://www.epa.gov/aqs), which shows ambient air quality data reported to U.S. EPA by the State of California. We then matched Network and regulatory monitors by reporting day and computed mean 24-h averages across all Network sites (*n* = 42) and regulatory sites in the Imperial Valley (*n* = 5 for PM_10_, and *n* = 3 for PM_2.5_—two of the PM_10_ sites do not measure PM_2.5_). We then computed seasonal and annual averages from the mean 24-h averages. Daily averages for Network monitors were calculated from hourly averages with 75% completeness (nine 5-min measures or more per hour). Only 24-h averages for Network monitors with 75% completeness (18 h or more per day) were included in this analysis.

To test for significant differences in annual averages between Network and regulatory monitors, we conducted two-sample *t*-tests for independence with unequal variance in OpenEpi, Version 3 (https://www.OpenEpi.com). Two-sided *p*-values indicated significance at the *α* = 0.05 level. We used restricted maximum likelihood estimation to calculate unbiased inter- and intra-monitor variance estimates from monitor-specific 24-h measurements. We ran ordinary least squares (OLS) simple linear regressions to estimate associations between mean 24-h averages from all Network monitors and mean 24-h averages from all regulatory monitors, by pollutant and year. To reduce bias in our estimates which were calculated from aggregated data, we used the bootstrapping approach with *n* = 1000 random samples from the observed data, with replacement. This method yields robust estimates of variance and bias-corrected 95% confidence intervals.

We computed Lin’s concordance correlation coefficient for agreement. This method combines measures of precision and accuracy to determine how far the observed data (measured by two different methods) deviate from the 45° line of perfect concordance. Data were processed and analyzed in R, Version 3.6.1 (R Foundation for Statistical Computing, Vienna, Austria) using *nmle*() (Pinheiro J, Bates D, DebRoy S, Sarkar D and R Core Team, 2016), *boot*() (Canty A, Ripley BD, 2020), *ggplot2()* (Wickham H, 2016), and *agRee()* (Feng D, 2020) packages for inter- and intra-monitor variance estimation, bootstrapping and regression plots, and concordance correlation coefficient estimation.

## 3. Results

### 3.1. Air Quality Results

After removing incomplete hourly and daily averages, there were 28,782 and 21,675 daily average values from 42 Network monitors between 29 May 2015 and 30 June 2018 for PM_2.5_ and PM_10_, respectively. In the same time period, there were 1767 PM_2.5_ daily averages from three regulatory monitors and 4858 PM_10_ daily averages from five regulatory monitors. After aggregating across monitor sites, we analyzed 1053 and 1051 paired Network-regulatory mean 24-h averages for PM_2.5_ and PM_10_, respectively.

Figure 1 shows median annual 24-h average PM_2.5_ and PM_10_ values from the Network during the 2015–2018 period. Median PM_2.5_ levels from the Network appear to be highest in winter, but there was no overall trend in increasing levels by time (Figure 1). Average 24-h PM_2.5_ values ranged from <1 to 255 μg/m^3^ during the time period. PM_10_ values also were highest in winter (Figure 1), except for Spring 2016 being the highest for that year. PM_10_ values ranged from 1.8 to 2431 μg/m^3^.

Table 1 shows average annual 24-h values of PM_2.5_ and PM_10_ from the Network and from CA regulatory sites in the Imperial Valley from 2015 to 2018 on a network-average basis. Mean values were not statistically significant for any year for PM_10_, but were statistically different for PM_2.5_ for 2015 and 2017. Data variability overall was much larger in the Network compared to the regulatory monitors, indicated by higher standard deviations, particularly for PM_10_. The 24-h average maximum levels of PM_10_ recorded by the community air Network reached 2430 μg/m^3^ in 2017 and the maximum level of 24-h mean of PM_2.5_ was 255 μg/m^3^ in 2018 (not shown).

Annual average PM_2.5_ measurements of the Network were consistently lower than those from regulatory monitors. These differences were only statistically significant in 2015 (*t* = 4.78, *p* < 0.0001) and 2017 (*t* = 3.56, *p* = 0.0004). This could possibly be explained by the fact that most of the regulatory monitors are located in more urban areas which would have higher levels of PM_2.5_, compared to the more diverse locations of the community monitors, or that the Dylos sensors are measuring PM_10_ more accurately than PM_2.5_. Community average annual levels of PM_10_ were not consistently higher or lower than the regulatory monitor average annual readings.

Monitor-specific 24-h averages were used to estimate variance between and within monitors for both Network and regulatory monitors, shown in Table 2. Inter-monitor variance was higher in the Network compared to that of regulatory monitors, suggesting greater variability across monitor locations in the Network. Intra-monitor variance was also higher in the Network, suggesting greater variability within each monitor location across time. Overall, intra-monitor variance was higher than inter-monitor variance for both Network and regulatory monitors, except for PM_2.5_ measurements from Network monitors which had greater variance between monitors than within.

Scatter plots and OLS linear regression lines for PM_2.5_ and PM_10_ are shown in Figure 2 and Figure 3, respectively. Network and regulatory mean daily averages were significantly associated for both PM_2.5_ and PM_10_ across all years. Linear associations were weak to moderate, as indicated by *r*-squared values, and robust estimates of standard error were low (Table 3). For PM_2.5_, regression lines show that Network monitors are, on average, predicting similar mean 24-h averages as the regulatory monitors. For PM_10_, regression lines show that predicted mean 24-h averages for Network monitors are similar to those of the regulatory monitors when estimates are less than 100 μg/m^3^. However, Network monitors are predicted to have higher mean 24-h averages for measures of PM_10_ in the range of 200–600 μg/m^3^ compared to predicted values for regulatory monitors.

Concordance coefficients indicate moderate agreement between Network and regulatory 24-h averages (Table 4). Agreement was similar for measurements of PM_10_ (Rho = 0.692, CI = 0.661, 0.720) and PM_2.5_ (Rho = 0.604, CI = 0.567, 0.638).

### 3.2. Under-Reporting by Government Monitors

In the beginning of 2016, the CA Air Resources Board began continuous PM_10_ monitoring at Calexico, CA in the Imperial Valley with a BAM FEM. The default setting of the BAM was capped at 985 μg/m^3^. Episodes examined in May of 2016 and October of 2017 showed while the FEM BAM showed values of “985,” the Dylos monitors were reporting readings exceeding 1800 and 1600 μg/m^3^, respectively [17]. Upon consultation with the U.S. EPA, CARB examined data from a collocated Dylos PM sensor from the Imperial Community Air Network for comparison. In December 2017, the FEM PM_10_ BAM was re-ranged by CARB to measure up to 4985 μg/m^3^. This correction will have an impact on historic and current average and maximum PM_10_ values.

### 3.3. Network Maintenance

Establishment of the Network has given the study partners the opportunity to learn and explore various issues with emerging low-cost sensor technology. The Imperial Community Air Monitoring Network has been running longer than other monitoring networks of its type, which are typically deployed for research purposes for a couple of months to a year. Through the study period, efforts were made to transfer knowledge, ownership and monitoring activities to CCV. All field maintenance tasks are currently managed by CCV staff, some who received training at the UW lab, and all hardware designs and software code were provided to CCV. CCV has begun deploying a complementary network of 17 meteorological monitoring stations, through an equipment loan from US EPA, which will offer real-time access to wind speed and direction data. Currently, each monitor is visited for maintenance every 45 days. The maintenance visits include a check-up of all hardware for wear and tear. Other activities outside of scheduled visits for reactive troubleshooting includes resetting the microcontroller and connecting the microcontroller back to the wireless network. In many locations, access to a reliable wireless connection is poor, so cellular hotspots had to be established. However, high temperatures in the region has led to mobile hotspot batteries becoming warped and battery replacement is sometimes necessary. CCV has found that batteries have to be replaced in high exposure areas once every 2 years.

The Dylos sensor has a factory established 2-year lifespan. Data are flagged when all bins read 0, when they jump up to an unreasonably high value and get stuck at that plateau, or the signal is slowly attenuated over time. When hardware failure occurs, the old sensor is switched out with a new unit. Although all the sensors from the original monitor deployment have been replaced, so far only a handful (5–6) have had to be replaced again for hardware issues. CCV has developed maintenance and troubleshooting records and an internal activity log which staff use to record any visits to the monitor sites, allowing staff to share common issues at each site with each other and the external QA/QC staff. Due to sensor drift and lifespan, factory recalibration is necessary.

## 4. Discussion

PM exposure is associated with a number of adverse health outcomes, including respiratory and cardiovascular disease, adverse reproductive outcomes, neurologic disease, and premature death [18,19,20,21]. Approximately 140,000 deaths per year were attributable to total PM in the U.S. from 2000 to 2010 [22]. Climate change is expected to increase wildfire risk, which, in turn, will increase particulate matter levels, and associated health risks [23]. Exposure to PM is not uniform among population groups; higher levels of PM have been found to be associated with higher deprivation indices and low economic position in a recent review [24].

Initial results from the Network from approximately 4 years of data shows that the Network is reporting more comparable data for PM_10_, than for PM_2.5_, in comparison to regulatory monitors. We found no statistically significant differences in the network-average means between the Network and regulatory systems for PM_10_. While OLS R^2^ values for PM_10_ ranged from 0.16 to 0.67, they showed improvement each year of operation of the Network. Average PM_2.5_ measurements from the Network were consistently lower than those from regulatory monitors. Network PM_2.5_ means differed from regulatory means by 6–17%, compared to 3–6% for PM_10_.

Overall, precision of the Network measurements was lower than the regulatory monitors, with coefficients of variation (CV) of the Network monitors ranging from 77% to 109% for PM_2.5_ and 95% to 166% for PM_10_; while the regulatory monitors had CVs ranging from 59% to 73% for PM_2.5_ and 67% to 88% for PM_10_. Further, we found that inter- and intra-monitor variance was higher in the Network compared to that of the regulatory monitors. Beyond higher sensor-to-sensor variability in the Network, lower precision of the Network compared to the regulatory monitors is not surprising, as the larger variation likely reflects true spatial variation in PM levels in the Valley, as the community Network monitors were distributed over a much larger region and at many more sites than the regulatory network. Although the sensors were calibrated to the original algorithm on installation, over time they have experienced sensor drift and some sensors have been replaced.

For the community monitors, annual averages of 24-h PM_2.5_ were in the 9–11 μg/m^3^ range, while 24-h PM_10_ annual averages ranged from approximately 45 to 56 μg/m^3^. The PM_2.5_ averages were below the CA standard of 12 μg/m^3^, but the PM_10_ annual averages were over twice the average annual average California standard of 20 μg/m^3^. The 24-h average maximum levels of coarse particulate matter (PM_10_) recorded by the community air Network reached over 2000 μg/m^3^ during the time period (over 40 times the maximum 24-h mean recommended by the World Health Organization (50 μg/m^3^; [25]), and the maximum level of 24-h mean fine particulate matter (PM_2.5_) was 255 μg/m^3^ (over 10 times the WHO maximum 24-h mean of 25 μg/m^3^). The 24-h average maximum level of over 2000 μg/m^3^ for coarse dust even exceeds measurements found during sand storms in Beijing [26]. However, the maximum levels measured by the Network should be interpreted with caution. The Dylos sensor was not calibrated for extreme high PM_10_ values as the reference monitors only reported values up to 985 μg/m^3^ for PM_10_ (and were capped at that level) and thus have not been validated at levels exceeding that limit. More calibration and validation work needs to be done to establish the Dylos upper-range in the Imperial Valley.

Previous work has compared Dylos or other light-scattering PM sensors to high-end instruments, or Federal Equivalent Method (FEM) or Federal Reference Method (FRM) monitors. These studies have selected different light-scattering instruments to evaluate, and different reference sources, making comparisons difficult due to potential instrument bias. Ioakimidis et al. [27] evaluated a Laser PM_2.5_ (Nova) using roadside measurements from a mobile laboratory. They used a high-end Optical Particle Sizer (TSI OPS 3330) (not FRM or FEM) for comparison. They found a R^2^ of 0.98 after adjusting for temperature and humidity. Castell et al. [28] evaluated 24 AQMesh units which measure total particle counts which are converted into PM mass-based fractions. Upon colocation with European Committee for Standardization reference analyzers for 5 weeks, they found an average r of 0.51. Budde et al. [29] evaluated a low-cost network in Germany using a light-scattering Sharp GP2Y1010 PM sensor and evaluated it against a EU PM reference monitor (Grimm Technologies Model EDM 180 PM Monitor) for 7 days. They found that the low-cost off the shelf sensor tracked the reference monitor well, but with a constant offset. No OLS regression results were reported. Zheng et al. [30] evaluated the Plantower sensor (Plantower model PMS3003, Plantower Technology, Beijing, China) in a low concentration setting in Research Triangle Park and a high concentration setting in urban India. They found R^2^’s of 0.66–0.95 depending on averaging time in the low concentration setting (compared to a Teledyne model T640 FEM) and R^2^s of 0.61–0.93 in the high concentration (compared to a E-BAM) depending on monsoon season and averaging time. Jiao et al. [31] evaluated the performance of the Community Air SensorNetwork (CAIRSENSE) project, a network of low-cost sensors in suburban areas of the Southeast U.S. For PM_2.5_, they compared two Dylos 1100 sensors, one with bin sizes ≥ 1 um (PC) and one with bin sizes ≥ 0.5 um (PC-PRO), to a MetOne BAM 1020 FEM PM_2.5_ monitor as reference. R^2^’s from OLS ranged from 0.33 to 0.45. For comparison, we found an *R*^2^ for converted hourly averaged Dylos mass measurements versus a PM_2.5_ BAM of 0.79 in our previous calibration work [6], and ranges of 0.35–0.49 for PM_2.5_ and 0.15–0.67 for PM_10_ in our post-implementation network-wide comparison reported here.

There are limitations and advantages when comparing low-cost sensor air monitoring networks to regulatory monitors. One main obvious advantage of low-cost sensor networks is the ability to greatly increase the spatial coverage of the domain of interest. In many areas, especially in more rural ones, the number of government monitors are sparse. Installation of additional government monitors happens slowly, due to the length of the siting process and cost. Government monitoring systems are primarily used to measure ambient background pollution levels for enforcement, while low-cost systems can be sited and put into operation more quickly. Low-cost systems can also be used as a check on regulatory monitors, as we have shown with the under-reporting of maximum values by the regulatory monitors in this study.

A low-cost system can detect many more elevated air pollution episodes than a regulatory network. In an analysis of the Network data from October 2016 to February 2017, 1426 exceedance episodes of PM_2.5_ were identified by the Network community monitors compared to only 116 identified by government monitors (over 12 times as many) [32]. Thus, more spatially refined air quality information can be used to help pinpoint pollution episodes important for adverse, acute exposures to particulate matter, individual exposure reduction, and reduce exposure misclassification. In addition, the Network data has been used for spatial and temporal modeling [33] and to model PM_2.5_ concentrations [34,35].

Previous work has shown that taking a community-based approach to air monitoring in this project increases local environmental health literacy, and provides direct benefits to community partners, such as engaging youth, and increased capacity and knowledge about air quality, data interpretation, and health impacts [36,37]. CCV led an extensive campaign to publicize the Network, presenting information during community events, meetings with school and other local officials, and interviews with news media. Members of the community steering committee also participated in this effort by sharing information about the Network, including presenting at schools, churches, and public meetings and by writing news articles. Data from the Network are also being used by schools and agencies. For example, with support from CCV, 10 schools with monitors have school flag programs which use real-time data to inform actions to reduce exposure, such as keeping students indoors when air quality is poor. The school uses an outdoor flag to communicate current air quality, changing its color several times a day to correspond to the community air levels (CALs) on the IVAN website (https://ivan-imperial.org/air). A senior center also established its own flag program to share data from its nearby community air monitor. Policy and public health actions, such as school flag programs and legislation providing resources for other communities to conduct similar projects, are also counted among the project results.

Approximately 3 years after the start of this project, CA Assembly Bill (AB) 617 was signed into law by Governor Brown on 27 July 2017. The law required the CA Air Resources Board, in consultation with local air districts, “to deploy community air monitoring systems, which shall be communities with high exposure burdens for toxic air contaminants and criteria air pollutants.” Subsequent to AB 617’s passage, AB 134 was passed, which institutionalized community air monitoring and provided up to $5,000,000 for technical assistance grants for community organizations to fund community air monitoring. Assembly member Eduardo Garcia, co-author of the bill, has stated that “AB 617 was modeled on the highly successful Imperial County Community Air Monitoring Network, which has demonstrated that empowering communities with the ability to monitor local air pollution can lead to key policy victories and improve public health.” (Eduardo Garcia, CA District 56, personal communication, 7.26.19)

There are also limitations of low-cost sensor networks. Low-cost networks need to be sustained financially, and staff need to be trained to troubleshoot hardware and software, and maintain data repositories. Sensors suffer from drift and short lifespan, and re-calibration or replacement is necessary. Low-cost systems, over time, produce millions of records. This also presents challenges for data visualization and interpretation for the public. As described above, we constructed a public website which displays and interprets the data, which was designed with public input to make sure the data were understandable by the community. Data from the Network is also available via an email alert system that notifies registered users when air quality is unhealthy.

## 5. Conclusions

In this paper, we presented the results of a community-based air monitoring Network which is providing real time, actionable data to neighborhoods affected by air pollution. We took a community science approach which is responsive to community concerns and scientific accuracy. A main hindrance to widespread adoption of low-cost air sensors is data quality. We calibrated our sensors to federal reference and federal equivalent monitors and provide ongoing maintenance to the Network. Analysis of the Network data shows that the Network monitors have less precision than the regulatory monitors. This is to be expected with low-cost sensors when compared to federal reference grade instruments which cost thousands of dollars more, prohibiting their widespread use.

In conclusion, we found in this study that the Network is reporting more comparable data for PM_10_, than for PM_2.5_, in comparison to regulatory monitors, and that the precision of the regulatory monitors, as expected, was higher than the Network monitors. On the other hand, we found that the Network reported more elevated air pollution episodes than the regulatory network, and that the regulatory network had been under-reporting particulate matter readings.

Future community-based networks should emphasize effective partnership communications, and ongoing capacity building by training local staff to maintain the network after initial funding is depleted. This study suggests that future air monitoring efforts in other areas worldwide can benefit by combining scientific criteria and procedures to ensure high data quality, such as rigorous monitor calibration and validation, with community priorities, such as the type of pollutants to be measured and monitor siting, to increase the ability of communities to affect public health policy.

## Figures and Tables

**Figure 1 sensors-20-03031-f001:**
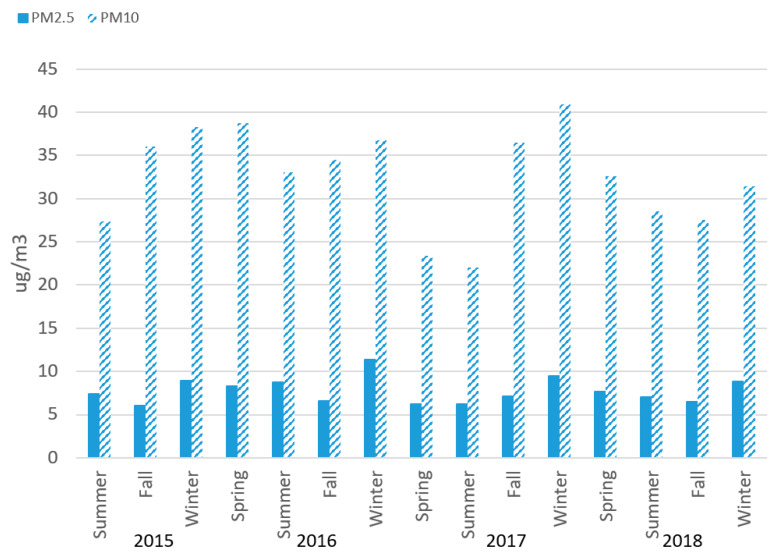
Median values (average 24-h measurements) of PM_2.5_ and PM_10_, Imperial Community Air Monitoring Network, 2015–2018, by season. Hours with less than 75% of completed 5-min measurements and days with less than 75% of completed 1-h measurements are excluded. Spring 2015 excluded due to low numbers of observations.

**Figure 2 sensors-20-03031-f002:**
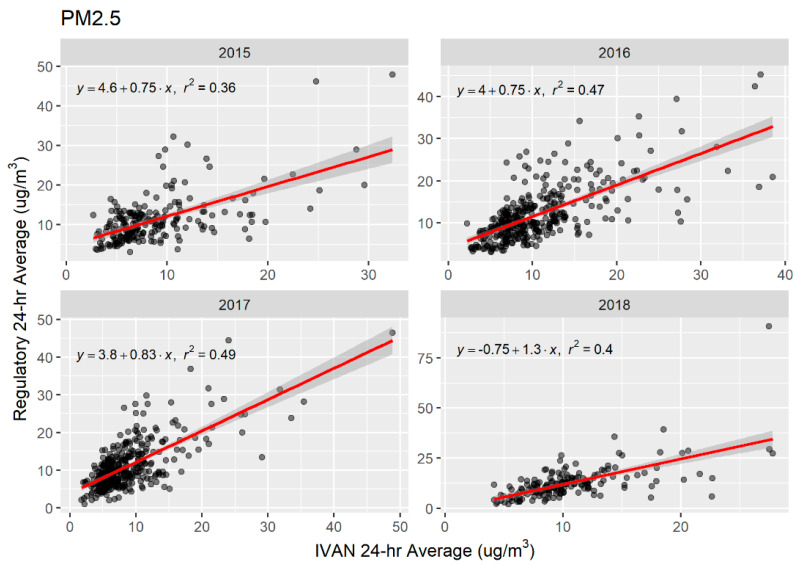
Scatter plots of mean 24-h averages of PM_2.5_ from Community Network (IVAN) and regulatory monitors, 2015–2018, by year. OLS simple linear regression lines shown in red with 95% confidence intervals (grey band).

**Figure 3 sensors-20-03031-f003:**
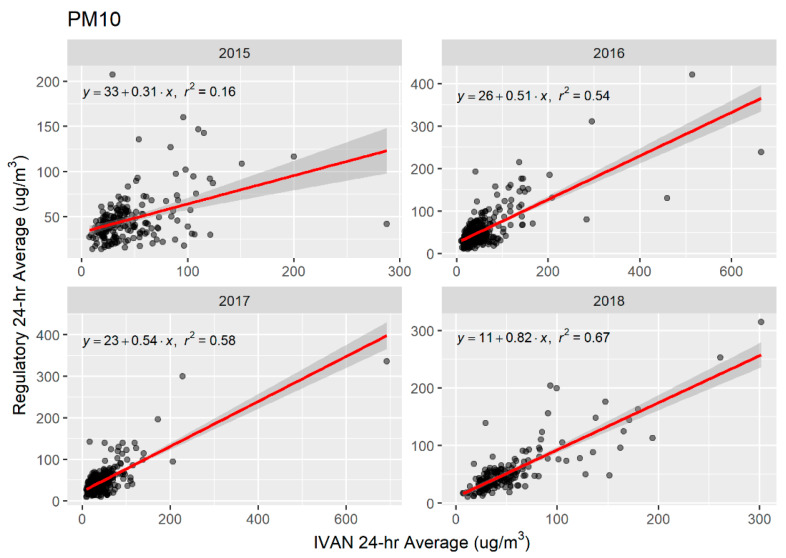
Scatter plots of average 24-h measurements of PM_10_, Imperial Community Air Monitoring Network and regulatory monitors, 2015–2018, by year. OLS simple linear regression lines shown in red with 95% confidence intervals (grey band).

**Table 1 sensors-20-03031-t001:** Annual particulate matter 24-h averages, standard deviations, coefficients of variation, two-sample *t*-test, and ordinary least squares (OLS) linear regression results. California Regulatory and Community Network Values, 2015–2018. Source: IVAN Network, U.S. EPA AWS.

		Regulatory ^†^			Community Network			Two-Sample *t*-Test *	
		Mean	SD	CV	Mean	SD	CV	*t* Statistic	*p*-Value
PM_2.5_	2015	10.7	6.4	59.4	8.9	6.8	77	4.78	<0.0001
	2016	11.7	6.9	58.9	11	8.8	80.4	1.88	0.06
	2017	10.7	6.5	61	9.2	10.1	109	3.56	0.0004
	2018	11.7	8.5	72.6	10.6	10.1	95.7	1.85	0.0674
PM_10_	2015	46	30.1	66.7	44.6	42.3	95	0.71	0.4763
	2016	53.6	47.3	88.2	55.2	83.8	151.6	−0.73	0.4631
	2017	45.5	36.4	80	42.8	71.1	166.1	1.52	0.1295
	2018	54.8	48	87.6	56	85	151.9	−0.40	0.6913

^†^ Regulatory data based on samplers using federal reference or equivalent methods. Extreme events are included. * Two-sample *t*-test for independence; two-sided *p*-values significant at the α = 0.05 level are bolded. Note: Only 6 months of data available for 2015 (Community Network data only available since 5/29/2015) and 2018 (regulatory data only available until 6/30/2018).

**Table 2 sensors-20-03031-t002:** Inter- and intra-monitor variance from restricted maximum likelihood estimation of 24-h average measures of PM_2.5_ and PM_10_ for Community Network (IVAN) and regulatory monitors.

		Inter-Monitor Variance	Intra-Monitor Variance
PM_2.5_	Community Network *	89.20	74.32
	Regulatory ^†^	3.86	45.98
PM_10_	Community Network *	251.67	5765.69
	Regulatory ^†^	36.09	1727.83

* Community Network (IVAN) includes 41 Dylos sensors located throughout Imperial County. ^†^ Regulatory monitors in Imperial County include three PM_2.5_ monitors in Brawley, El Centro and Calexico and five PM_10_ monitors in Niland, Westmorland, Brawley, El Centro, and Calexico.

**Table 3 sensors-20-03031-t003:** Bootstrapped estimates of associations between Network and regulatory mean 24-h average PM_2.5_ and PM_10_ measurements from ordinary least squares linear regressions.

		r^2^	95% BCa CI ^†^	Bias	SE *
PM_2.5_	2015	0.357	(0.204, 0.520)	−0.0018	0.081
	2016	0.468	(0.370, 0.592)	0.0010	0.057
	2017	0.489	(0.376, 0.612)	−0.0027	0.060
	2018	0.401	(0.233, 0.507)	0.0130	0.067
PM_10_	2015	0.157	(0.040, 0.322)	0.0197	0.080
	2016	0.538	(0.373, 0.739)	0.0019	0.093
	2017	0.577	(0.369, 0.762)	−0.0075	0.099
	2018	0.673	(0.469, 0.832)	−0.0105	0.094

^†^ 95% bias-corrected and accelerated bootstrap confidence intervals. * SE: bootstrap standard error.

**Table 4 sensors-20-03031-t004:** Calculation of Lin’s concordance correlation coefficient (Rho) to estimate agreement between Network and regulatory 24-h average PM_2.5_ and PM_10_ measurements.

	Rho	95% CI ^†^	b *
PM_2.5_	0.604	(0.567, 0.638)	0.926
PM_10_	0.692	(0.661, 0.720)	0.961

^†^ 95% confidence interval lower and upper limits. * Bias correction factor measuring how far the best-fit line deviates from a line at 45°. No deviation from the 45° line occurs when b = 1.

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
