# Peer review of "Performance of a Low-Cost Sensor Community Air Monitoring Network in Imperial County, CA"

_sensors, 2020, doi:10.3390/s20113031_

Round 1

Reviewer 1 Report

The authors addressed my comments. I'm fine with this version.

Author Response

We thank the reviewer for their helpful comments and feel that the manuscript is much improved.

Reviewer 2 Report

In this work, the authors evaluated the performance of a low cost air quality (particular matter) monitoring Network installed in Imperial County (40 sensors), which is delivering real-time data to local communities on levels of particulate matter to date by comparing the low-cost sensor readings to regulatory monitors for four years of operation (2015-2018) on a network/domain-wide basis.

It is very original and very interesting. Before the publication, some aspects need minor improvements such as:

  • Abstract: the abstract is too long and difficult to understand for the reader. The authors have to re-write it (max 200 words).
  • Introduction: in the paper there is not an analysis on the global effect of the pollution mitigation also through the implementation of low cost air quality network;
  • Materials and methods: the authors should describe as the air quality network was built. Is there a logic to install the 40 sensors? The authors can cite the paper on this: Sofia D, Lotrecchiano N, Giuliano A, Barletta D, Poletto M. Optimization of number and location of sampling points of an air quality monitoring network in an urban contest. Chem Eng Trans 2019;74:277–82. doi:10.3303/CET1974047;
  • Results: The theoretical assessment of the impact of the future diffusion of similar low cost air quality monitoring networks on world scale (e.g. metropolitan cities) was not consider. How the work results can improve the air quality monitoring of the future? The authors should touch on to this topic in the paper;
  • Results: at lines 241-243 the font is wrong;
  • Conclusions: Summary conclusions have to be added in the end of the paper.

Author Response

We thank the reviewer for their helpful comments and reply to each one in turn:

Comment #1: Abstract: the abstract is too long and difficult to understand for the reader. The authors have to re-write it (max 200 words).

Response: We have rewritten the abstract to make it easier to understand and it is now 200 words.

Comment #2: Introduction: in the paper there is not an analysis on the global effect of the pollution mitigation also through the implementation of low cost air quality network;

Response: We have added the following sentence in the introduction: “With the stall of international agreements to decrease emissions to address climate change, local community networks could result in cost-effective improvements to air monitoring and emission reduction at the local scale in the near-term and have policy relevance and positive developments globally for climate change and air pollution in the long-term.”

Comment #3: Materials and methods: the authors should describe as the air quality network was built. Is there a logic to install the 40 sensors?

Response: We have already published an article discussing the rationale on how the network was built and number of monitors selected. This was already mentioned in the methods on lines 88-91: “The rationale and siting for the 40 sensors were based on a two stage process where the community selected the locations of the first 20 sensors primarily based on locations of vulnerable populations and the second 20 based on specifications to meet the spatial requirements for land use regression [6].”

Comment #4: Results: The theoretical assessment of the impact of the future diffusion of similar low cost air quality monitoring networks on world scale (e.g. metropolitan cities) was not consider. How the work results can improve the air quality monitoring of the future? The authors should touch on to this topic in the paper;

Response: To respond to this comment, we have inserted the following sentence at the end of the discussion (lines 418-421) : “This study suggests that future air monitoring efforts in other areas worldwide can benefit by combining scientific criteria and procedures to ensure high data quality, such as rigorous monitor calibration and validation, with community priorities, such as the type of pollutants to be measured and monitor siting, to increase the ability of communities to affect public health policy.”

Comment #4: Results: at lines 241-243 the font is wrong;

Response: Fixed

Comment #5: Conclusions: Summary conclusions have to be added in the end of the paper.

Response: We added summary conclusions at lines 416-420 in the discussion.

Reviewer 3 Report

In the introduction there aren’t similar research cases and nothing literature work is cited. Similar cases can be from works:

  • Sofia D, Giuliano A, Gioiella F. Air Quality Monitoring Network for Tracking Pollutants: the Case Study of Salerno City Center. Chem Eng Trans 2018;68:67–72. doi:10.3303/CET1867052.

A figure with the map of sensors distribution could be added to better understand the positioning in the region Imperial County.

Author Response

We thank the reviewer for their helpful comments and respond to each in turn:

Comment #1: In the introduction there aren’t similar research cases and nothing literature work is cited. Similar cases can be from works:

Sofia D, Giuliano A, Gioiella F. Air Quality Monitoring Network for Tracking Pollutants: the Case Study of Salerno City Center. Chem Eng Trans 2018;68:67–72. doi:10.3303/CET1867052.

Response: We have added a section discussing the literature and referenced the above study in lines 30-35.

Comment #2: A figure with the map of sensors distribution could be added to better understand the positioning in the region Imperial County.

Response: We have previously published such a map in one of our earlier referenced publications, but will defer to the editor as they may not be allowed due to previous publication.

This manuscript is a resubmission of an earlier submission. The following is a list of the peer review reports and author responses from that submission.

Round 1

Reviewer 1 Report

This study reports on the performance of a particulate matter (PM) monitoring network that consists of low-cost real-time particle counters. This research topic meets the need for citizen science and community-based air pollution studies.

There are three major issues that the authors should address:

First, the performance was evaluated by comparing the results from the network with those by federal reference methods (FRMs). This is the core part of this study; however, the comparison was insufficient. The authors need to consider and clarify two questions for this purpose:

  • What concentrations are compared? By the averaging time, did you compare the annual average, seasonal average, monthly average, or daily average? By space, did you compare individual sensors, network-average, or an area surrounding an FRM monitor (e.g., within 2 miles)?
  • How were the two methods compared? There are many metrics and methods for the comparison purpose. EPA has simple metrics:

There are simple statistical methods, such as t-tests and simple linear regression by plotting new measurements against the reference measurements (X-Y plots).

There are complex statistical methods, e.g., Lin’s concordance correlation coefficient can evaluate accuracy and precision simultaneously. Check this review paper:

Theriogenology

Volume 73, Issue 9, June 2010, Pages 1167-1179

Method agreement analysis: A review of correct methodology

P.F.Watson and A.Petrie

Second, one valuable feature of the continuous network is to identify air pollution episodes. To demo this application, the authors should present figures that display peaks of PM concentrations and indicate the episodes.

Third, this reviewer suggests that the authors acknowledge the limitations and disadvantages of using air pollution sensors from their long-term experience with this network. EPA’s evaluation of low-cost sensors pointed out that none of the evaluated sensors provided accurate measurements, including the Dylos monitors. The maintenance and durability of sensors are still questionable. The overall costs are not low, and program sustainability is difficult. The real-time sensors generate tons of measurements, which pose huge challenges to data validation, summarization, and presentation for the general public.

Author Response

We thank reviewer #1 for their comments and we feel their comments have substantially improved the paper.  We respond to each of their comments in turn:

1) What concentrations are compared? By the averaging time, did you compare the annual average, seasonal average, monthly average, or daily average? By space, did you compare individual sensors, network-average, or an area surrounding an FRM monitor (e.g., within 2 miles)?

Response: Concentrations are compared in ug/m3 from both the Dylos sensors (after conversion of counts to mass) and from the federal reference monitors (US EPA).  The Network and regulatory monitors were matched by day and then daily averages were computed.  Then annual averages were computed from the daily averages.  This is already described at the end of the methods section.  We used a network average for comparison, as the domain of monitoring for the Network and the regulatory monitoring are the same.  We added a sentence on this at the end of the methods.

2) How were the two methods compared?

Response:  We thank the reviewer for their suggestions on this.  We have now calculated t-tests for the differences in means, have run OLS regression and reported R2's and have computed Lin's concordance correlation coefficient.  Results of these analyses are in tables and in the results section.

3)  one valuable feature of the continuous network is to identify air pollution episodes. To demo this application, the authors should present figures that display peaks of PM concentrations and indicate the episodes

Response: This work and figures have been previously published.  We describe these findings briefly and reference in the discussion (lines 320-325 )

4) this reviewer suggests that the authors acknowledge the limitations and disadvantages of using air pollution sensors from their long-term experience with this network.

Response:  We agree and have added a paragraph addressing this in the discussion (lines 310-365)

Reviewer 2 Report

Comments

  • Line 61: The Network is of (add)
  • Line 127: 2018 instead of 2015?
  • Line 143-145: The authors claim that ‘This could possibly be explained by the fact that most of the regulatory monitors are located in more urban areas which would have higher levels of PM2.5, compared to the more diverse locations of the community monitors’.

Although this might be possibly true explained by the use of median values in Figs 1-2, still cannot be proved by the findings since it is not referred anywhere in the paper that the Community sensors devices were: a) either measure directly with the existing fixed regulatory monitoring stations from US EPA in the same location and period of time, b) 40 sensors were either enough? (or not)? to give the same information and under the same spatial area monitoring as the ones used by US EPA? Would it be the same with more or less sensors? And how did they categorize the places to install the sensors? Therefore it is asked by the authors to further clarify and explain this part while giving more information.

  • It is indeed up to now accepted in the scientific community that the more expensive static/fix monitoring/regulatory systems used by the various national agencies on air pollution do offer accurate results compared to the low cost sensors technology, still is this something that the authors argue/agree (or not) and if so then would rather show it with a number of refs (papers) from existing wide literature (a couple below).
  1. a) K. N. Genikomsakis, F. Galatoulas, P. Dallas, L. M. C. Ibarra, D. Margaritis, C. S. Ioakimidis, ‘Development and on-Field Calibration of Low-Cost Portable System for Monitoring PM2.5 Concentrations for Smart Cities application’, DOI: 10.3390/s18041056, 18, 1056, Sensors, (2018).
  2. b) Nuria Castell, Franck R.Dauge, Philipp Schneider Matthias Vogt Uri Lerner Barak Fishbain David Broday, Alena Bartonova ‘Can commercial low-cost sensor platforms contribute to air quality monitoring and exposure estimates?’ Environment Internaitonal, Volume 99, February 2017, Pages 293-302.
  • Lines 240-257 seems to have a larger space between them compared with rest of text, please check.

Thus for all the above the reviewer suggests a major review.

Author Response

We thank reviewer #2 for their helpful comments and feel the manuscript is much improved after addressing their comments.  Below are their comments and our response in turn:

1) Line 61: The Network is of (add)

Response: Corrected

2) Line 127: 2018 instead of 2015?

Response:  Actually, 2015 is correct.  As can be seen in Fig. 1, Spring 2015 is omitted due to low numbers, but Spring 2018 is included in the figure.

3) The authors claim that the Network PM2.5 measures may be lower than the regulatory monitors due to the fact that regulatory monitors are located in more urban locations than the Network. This statement cannot be proved unless the Network sensors measure directly with the regulatory monitoring stations which is not referenced.

Response: Although we were stating this as conjecture, not fact, the reviewer appears to be asking if the sensors were ever calibrated to the regulatory monitors.  We in fact refer to the previously published calibration results in the second para of the introduction, and the fourth paragraph of the methods.  We also have added the results of the calibration in the discussion with the literature review, "For comparison, we found an R2 for converted hourly averaged Dylos mass measurements versus a PM2.5 BAM of 0.79 in our previous calibration work."

4) In reference to the above question, are 40 sensors enough or not to give the same information and under the same spatial area monitoring as the US EPA monitors?

Response: The rationale and siting for the 40 sensors were based on a two stage process where the community selected the locations of the first 20 sensors primarily based on locations of vulnerable populations and the second 20 based on specifications to meet the spatial requirements for land use regression.  The number of sensors were also limited by cost.  This is all previously published work.  We have inserted a sentence in the methods referring to the previously published work.  We would assert that adding more sensors would decrease the variability and increase the precision of the Network when comparing to the regulatory data.

5)  how did they categorize the places to install the sensors?

Response: This has been previously published.  We had already referred to this in the methods: 

"We developed a process to involve community residents to select monitoring sites, which involved the use of mobile devices equipped with a custom-designed mobile web form, modified from IVAN Imperial (https://ivan-imperial.org/air), CCV’s existing community environmental reporting website [6]. Residents used government-based monitor siting criteria (such as building height, security, likelihood of available Wi-Fi and AC power supply, and locations of nearby air pollution sources) in their assessment."

6) Do the authors agree that the more expensive static/fix monitoring/regulatory systems used by the various national agencies on air pollution do offer accurate results compared to the low cost sensors technology?

Response: We now include a discussion of the advantages and disadvantages of the low cost network compared to a regulatory network in the discussion.

7) Lines 240-257 seems to have a larger space between them compared with rest of text, please check.

Response: Done

Reviewer 3 Report

1. Air quality-related work did attract a lot of attention recently. This paper presents the results of a community-based air monitoring Network, which is providing real-time air quality data.   2. The introduction does not specify the main contribution of the paper. The objectives and the context or motivation of the research are not described. The problem statement is not clearly stated in this section.    3. The literature review is missing. The related work must be described, and the research gap must be clearly stated.    4.  The specifications for various sensors used need to be presented listed in detail. Moreover, no experimental data results are presented for power consumption, processing/transmission time, etc., for the system since these data are essential for further design or usage considerations and also because of the scope of the journal.   5. The results must be compared with similar approaches using a table to clearly state the gap and highlight the contribution of your work.   Moreover, this paper can not be considered as a research article in this present form. The paper does not state clearly the outcomes and limitations of the study. In sum, this paper needs a total revision, which can not be done via major revision. Therefore, I recommend rejecting this paper.      

Author Response

We thank reviewer #3 for their helpful comments and feel the manuscript is much improved after addressing their comments.  Below are their comments and our response in turn:

1) The introduction does not specify the main contribution of the paper.  The objectives and the context or motivation of the research are not described.  The problem statement is not clearly stated in this section.

Response: We have revised the introduction to highlight the contribution, objectives, problem statement, and context/motivation of the research, including that there have been few performance evaluations of low-cost networks operating in the field, none which have reported with such extensive data.  No studies have been conducted evaluating performance of a low-cost network post implementation on a network-wide basis, compared to regulatory data.

2) The literature review is missing.

Response:  We now present a more thorough literature review in the discussion.

3) Specifications of the sensors should be presented in detail, including power consumption, processing time...

Response:  The specifications of the Dylos sensor have been previously published.  We have added details on power consumption and processing time and the previous work is clearly referenced.

4) Results must be compared with similar approaches in a table to clearly state the gap and highlight the contribution of the work.

Response:  Our results are not directly comparable to other research as the previous work is essentially reports of results of collocation and calibration experiments.  To our knowledge, no other research has evaluated a low-cost air monitoring network post-implementation on a network-wide basis.  However, we do report R2's of previous research using similar sensors for comparison to the R2's we report here and in our previous calibration research.

Round 2

Reviewer 1 Report

The authors clarified that they compared the annual means measured by the sensor network and those by the EPA reference methods.

From this reviewer’s understanding, the values of low-cost sensors are:

  • They can be easily deployed to form a network;
  • The sensors provide continuous readings, meaning measurements with high temporal resolutions;
  • The network can show the spatial variability of PM concentrations.

One valuable application is to identify air pollution episodes, which, however, has been reported in the authors’ previous publication. Then the value of this paper is to show how the sensor network can provide more information on spatial and temporal variability than the FRM monitors.

The current validation of the network performance was only for the annual means. This overall theme missed the core values of the sensor network.

Regarding the statistical details, the comparisons had many problems. It was not clear how the authors made multi-sensors to one FRM monitor comparisons. Did the comparisons consider intra- and inter-sensor variations? How did the comparisons account for repeated and paired measurements?

I feel the data collected are valuable, and I expect many citizen science projects will cite this study, if published. Thus the authors should make solid conclusions. However, the statistical analyses in this manuscript are weak and may give misleading conclusions. This manuscript is not up the standards for publication in its current format; however, if the authors design a clear comparison framework and use correct statistical methods, this study may be publishable.

Reviewer 2 Report

The authors have responded adequately and satisfactorily in my suggestions, therefore I recommend it to be accepted as it is. 

Reviewer 3 Report

I am not satisfied with the author answers, particularly on point 4).